# The Effect of Role-Play on the Development of Dialogue Skills among Learners of Arabic as a Second Language

**Ayman Sabry Daif-Allah** [1,*] and **Muhammad Sultan Al-Sultan** [2]

1   Department of English Language and Translation, College of Arabic Language and Social Studies, Qassim University, Buraydah 52571, Saudi Arabia
2   Department of Arabic Language and Arts, College of Arabic Language and Social Studies, Qassim University, Buraydah 52571, Saudi Arabia
*   Correspondence: abdalaal@qu.edu.sa

**Abstract:** Dialogue is one of the most important methods of achieving a desired goal and persuading others to accept a certain idea. This research aims to identify the effect of the role-play strategy on the development of dialogue skills among advanced-level students in the Arabic Language Teaching Unit for Non-Native Speakers at Qassim University. Using the quasi-experimental approach, the study was administered to 50 students, 25 students in the experimental group and the same number in the control group. The experimental group was taught through role-play, while the control group was taught through the traditional textbook-based methods. Quantitative data were collected by means of a dialogue skills assessment scale designed to assess the designated dialogue skills of the study sample. A quantitative analysis of the results showed a positive effect of role-play on the development of dialogue skills in terms of self-esteem, good listening, the expression of opinion, and respect for others. The results also showed that the simulation of real-life situations motivated students to develop dialogue skills in an organized manner. The results also highlighted the positive and negative aspects of using role-play in enhancing dialogue skills of learners of Arabic as a second language. The study recommends expanding the use of the role-play strategy in teaching the various basic skills of the Arabic language to the non-native speakers at Qassim University. It also suggests useful ideas for other researchers to study innovative techniques for improving the acquisition of language skills among learners of Arabic as a second language.

**Keywords:** role-play strategy; dialogue skills; Qassim University; teaching language; non-native speakers

## 1. Introduction

Arabic is the only Semitic language that has been preserved, has not changed in features, and will not become extinct, as languages tend to do, due to the revelation of the Holy Qur'an in a clear Arabic tongue. Successful learning of Arabic is associated with the Noble Qur'an because it is not possible to appreciate the miracles of the Holy Qur'an and understand it accurately in a language other than Arabic. Hence, learning Arabic is an aspiration for any Muslim, regardless of his color or race. Arabic has also been approved as one of the six official languages of the United Nations, and UN organizations and many non-Arabs and non-Muslims are eager to learn Arabic to know about the Arabic heritage and Islamic civilization. Today, as interest in studying the Arabic language is increasing, it is necessary for Arabic language instructors to be proficient in the concepts of specialization so that they can effectively teach Arabic to a wider audience [1]. This goal will be difficult to achieve without a thorough knowledge of various innovative teaching strategies.

Role-play enhances learners' values and thinking, problem-solving, and dialogue abilities. It is defined as a specific performance that maximizes the roles of learners and supports them directly and indirectly in learning the language. It helps students acquire skills, knowledge, and experiences in a relaxed learning environment characterized by

freedom and pleasure [2]. Dialogue is a type of communication between two parties on a particular issue of thought, science, and knowledge. It takes place in an equal and comfortable manner and is characterized by tolerance and avoidance of disagreements. People use a variety of dialogue skills to show the truth through appropriate presentation of argument, offering proof of thoughts and acceptance of the ideas of others [3,4]. Dialogue is a significant method through which different people linked by common interests can reach an understanding. These people can understand the reasons underlying their behaviors and choose appropriate topics for carrying out friendly dialogues [5]. In this study, dialogue refers to communication skills of people that enable them to effectively cooperate and interact with other people verbally and non-verbally, while preserving self-esteem, respecting others, expressing opinions, and showing good listening skills.

The results of a pilot study administered by the authors revealed weaknesses in the language of dialogue used inside classrooms between learners and instructors on the one hand and between learners and their colleagues on the other. Participants in the pilot study, including students and faculty, attributed the reason of the learners' poor dialogue skills to the use of traditional teacher-centered teaching methods in the classroom. In addition, assessment of dialogue activities focused on assessing the skills of accuracy, fluency, ideas, and style and neglected other important basic skills, such as preserving self-esteem, respecting others, expressing opinions, and displaying good listening skills. This study assumes that role-play would be important in developing these skills. This presupposition is confirmed by many studies in the field of teaching Arabic and English to non-native speakers. The use of appropriate strategies has a positive impact on dialogue skills development and provides learners with the opportunity to practice the correct language in different real-life situations [3,4,6]. The positive effect of the role-play strategy in developing various communicative skills has been widely discussed in the literature. However, there are few studies at the regional and local levels that deal with the use of role-play in developing dialogue skills among learners of Arabic as a second language. Therefore, the authors felt the need to introduce the role-play learning strategy to help non-native speakers of Arabic develop their dialogue skills in all its aspects as it allows them a great deal of freedom in presenting and discussing their ideas in an enjoyable, relaxed environment. Accordingly, this study seeks to answer the following questions:

1. How can the role-play strategy be used in developing dialogue skills in a meaningful context among the advanced-level students in the Arabic Language Teaching Unit for Non-Native Speakers at Qassim University?
2. Are there any statistically significant differences between the scores of the experimental and control groups in the dialogue skill test due to the role-play strategy?
3. What are the positive effects of the role-play strategy on dialogue skills in relation to self-esteem, respect for others, the expression of opinion, and good listening?

*Purpose of the Study*

This study aims to identify the effect of role-play on the development of dialogue skills among advanced-level students of Arabic as a second language in the first semester of the academic year 2022/2023 in the Arabic Language Teaching Unit for Non-Native Speakers at Qassim University via the following steps:

1. Identifying the dialogue skills of level-six students of Arabic as a second language in the Arabic Language Teaching Unit for Non-Native Speakers at Qassim University.
2. Recognizing the effect of the role-play strategy in the development of dialogue skills among advanced-level students of Arabic as a second language in the Arabic Language Teaching Unit for Non-Native Speakers at Qassim University.
3. Identifying the differences in the cognitive domain between the average scores of the students in the experimental group (who developed dialogue skills through the role-play strategy) and the control group.

4.  Presenting one of the active teaching strategies as a model for teaching different language skills in the Arabic Language Teaching Unit for Non-Native Speakers at Qassim University.
5.  Attracting the attention of the senior leaders at the Arabic Language Teaching Unit for Non-Native Speakers at Qassim University to the importance of the role-play strategy in developing students' cognitive aspects and providing the students with an opportunity for displaying creativity, innovation, and initiative.

## 2. Review of the Literature

One of the main objectives of teaching Arabic as a second language in the Kingdom of Saudi Arabia is to enable students to communicate effectively in Arabic in various academic, social, and daily-life situations [7]. One possible way to achieve this end is helping learners develop basic dialogue skills that help them communicate effectively with one another through a thorough understanding of the nature, objectives, and etiquette of dialogue between two conflicting parties [4]. Due to the great importance of dialogue for educational institutions, most countries have been interested in spreading and developing a culture of dialogue through various programs and events at various levels [7]. Many centers and associations have been established and many dialogue forums have been designed on the Internet that hold conferences and seminars aimed at spreading and developing a culture of dialogue among all members of society [8]. In the educational field, dialogue contributes to raising the level of students' thinking and helps them to form their ideas, strengthen their knowledge, and increase their ability to understand and think. In addition, involving students in dialogue situations stimulates their critical thinking and provides them an opportunity for scientific growth and developing personality based on knowledge [9].

Therefore, educators aspire to develop dialogue principles among individuals in society using innovative teaching strategies. Role-play is one of the most widely used teaching strategies that help learners develop dialogue skills and linguistic expression and interact with others [2]. In an educational setting, the student assumes one of the roles that exist in real life and interacts with others within the limits of this role. This method draws and retains learners' attention and motivates them to learn [10]. It is, therefore, considered effective in helping students understand themselves and others as it also involves creating a more positive and lively interaction in the classroom [1].

Several studies investigating the impact of role-play on teaching languages as a second language have found that role-play (1) creates a spirit of vitality and fun in the educational situation, (2) contributes to strengthening positive communication among students and developing social spirit, intimacy, and love among them, (3) helps discover students with outstanding and high abilities, (4) helps learners use their experiences in real-life situations, (5) helps learners develop thinking and an analytical mind, (6) creates an opportunity for practice, training, and feedback, (7) provides the student direct teaching experience, (8) and develops independence, self-confidence, and pride among learners [11–14]. However, this strategy can be time-consuming, and the behaviors of some learners may be illogical and dissociated from reality. In addition, this method may not suit shy learners and may cause embarrassment and anxiety among those who have not mastered the art of acting. It also needs constant teacher supervision, and role-play may become a form of entertainment rather than a learning opportunity [15].

Given the importance of role-play in enhancing dialogue skills, the literature is rich with studies that have examined the culture of dialogue and the acceptance of others in different parts of the world. Findings of related studies have shown that dialogue skills are of great importance in learning a language. Instructors are always looking for new ways of helping learners develop skills to express their opinions and support their arguments in a peaceful and comfortable manner [16–22]. In addition, recent research has shown that students improve in terms of language fluency and dialogue skills by using body language during face-to-face communication when they have the opportunity to participate in role-play [3,14,23]. The role-play strategy is effective in developing many

aspects of dialogue, such as self-confidence, ethics of conduct, and good listening skills, as well as values, feelings, and attitudes of the person himself. It also enables students to think by listening to the ideas of others and to clarify, expand, and discuss those ideas as well as to present new ideas [24,25]. However, previous research has not addressed the development of dialogue skills among learners of Arabic as a second language at Qassim University. In addition, related studies have focused on the development of the dialogue skills in terms of accuracy, fluency of ideas, and style. This study, therefore, intends to fill a gap in research by investigating the effect of role-play on improving four specific dialogue skills, i.e., preserving self-esteem, displaying good listening skills, expressing opinions, and respecting others, among non-native speakers studying Arabic as a second language.

In addition, most related studies have used descriptive approaches, involving questionnaires or interviews, which are the most widely used methods. However, this study is significant as it uses the quasi-experimental approach, a most significant approach to investigating educational issues that combines the concepts of dialogue culture and the acceptance of the other in real classroom active learning.

## 3. Materials and Methods

### 3.1. Research Design

The quasi-experimental research design was used since it provides detailed information on the impact of the independent variable, the role-play strategy, on the dependent variable, dialogue skills, for advanced-level learners of Arabic as a second language at Qassim University.

### 3.2. Participants

The study involved 50 advanced-level male students studying Arabic as a second language in the first semester of the academic year 1444 AH at the Arabic Language Teaching Unit for Non-Native Speakers at Qassim University. These students were divided randomly into two groups, 25 students each, one representing the experimental group and the other representing the control. Their mean ages ranged from 23 to 26 years old, and they came from 10 different countries. The academic content provided to both groups was identical. However, the experimental group studied by using the role-play strategy, while the control group studied by the traditional methods based on the textbook.

### 3.3. Instruments

This study aimed to identify the effect of role-play on the development of dialogue skills among advanced-level students of Arabic as a second language in the first semester of the academic year 2022/2023 at the Arabic Language Teaching Unit for Non-Native Speakers at Qassim University. Toward this end, a dialogue skills assessment scale was designed to assess the effect of role-play on the development of dialogue skills among advanced-level students studying Arabic as a second language at Qassim University. This scale was developed and adapted from Idham (2022), Al-Harahsheh (2017), and Huba and Freed (2000). The scale consisted of four standards, five indicators each, that measure the specific dialogue skills. As Table 1 shows, 1 mark was assigned to each indicator, and the maximum score of the four standards was 20 marks.

The appropriateness of the dialogue skills assessment scale was verified by conducting a pilot study among 10 students and presenting the results to a jury committee of five professors specialized in teaching Arabic as a second language. Feedback from the pre-administration and the jury committee showed the appropriateness of the scale to the research aims. In addition, a Cronbach alpha score of 0.73 showed acceptable consistency of reliability.

**Table 1.** Standards of the Dialogue Skills Assessment Scale.

| No | Standards | Indicators | Maximum Score | Percentage |
|---|---|---|---|---|
| 1 | Self-esteem | The interlocutor shows confidence in his abilities. | 1 | 5% |
| | | The interlocutor does not take much time to make a decision. | 1 | 5% |
| | | The interlocutor cares about his appearance. | 1 | 5% |
| | | The interlocutor does not confuse thoughts. | 1 | 5% |
| | | The interlocutor does not give up easily. | 1 | 5% |
| 2 | Good listening | The interlocutor shows interest in what other people say by using phrases such as yes, good, and true. | 1 | 5% |
| | | The interlocutor requests clarification of the content of the message instead of guessing. | 1 | 5% |
| | | The interlocutor asks for clarification to understand others' point of view. | 1 | 5% |
| | | The interlocutor does not interrupt or respond to the speaker before the speaker finishes speaking. | 1 | 5% |
| | | The interlocutor takes notes of what is important in the opponent's speech. | 1 | 5% |
| 3 | Expression of opinion | The interlocutor expresses his opinion in a set of arranged points. | 1 | 5% |
| | | The interlocutor provides logical evidence that supports his point of view. | 1 | 5% |
| | | The interlocutor adheres to the time specified for expressing his ideas. | 1 | 5% |
| | | The interlocutor tolerates opposing points of view. | 1 | 5% |
| | | The interlocutor exchanges opinions and ideas with the opponent in order to resolve the contentious point. | 1 | 5% |
| 4 | Respect for others | The interlocutor begins the debate by greeting the opponent with kindness and affection. | 1 | 5% |
| | | The interlocutor shows respect for the speech and behaviors of others and does not mock them. | 1 | 5% |
| | | The interlocutor accepts other people's hobbies, talents, and interests and does not make fun of them. | 1 | 5% |
| | | The interlocutor avoids psychological tension and verbal violence. | 1 | 5% |
| | | The interlocutor shows respect for the opinion of the opponent even if it contradicts his own opinion. | 1 | 5% |
| | Total | | 20 | 100% |

### 3.4. Process of the Experiment

The experiment was carefully planned and organized to ensure smooth implementation. The authors planned two orientation sessions of 90 min each. The aim of the first session was to develop among the participants in the experimental group awareness of the significance of developing dialogue skills as a necessary criterion for enhancing their motivation to acquire the values, etiquette, and skills of dialogue. This orientation session included presentation of short videos of effective and ineffective debates. The authors listened to participant comments on the videos and provided suggestions for carrying out a successful dialogue. During the second session, the authors discussed and presented the different common daily life roles assigned to the participants. The aim of this session was to obtain feedback from students and to ensure that the roles were appropriate in terms of the students' abilities and interests. Some roles that were agreed on were debates between father and son, salesman and customer, husband and wife, fast driver and police officer, instructor and student, school principal and parent, child and mother, patient and physician, grandfather and grandchild, and mother-in-law and daughter-in-law. Then, in the Dialogue Corner of the Multimedia Hall, students started practicing dialogue skills in real-life situations using the role-play strategy.

The participants were divided into five groups and met for 90 min twice a week for a total period of 10 weeks. The students would sit at a round table for dialogue on a pre-planned topic. Participants in each group would play the roles of both the proponent and the opponent by turn. At the beginning of each session, before the interlocutors started role-playing, the instructor would introduce each session with an overview to the debated topic. Other students of the experimental group would observe and take notes

without interfering. The role of the teacher was limited to observing the flow of dialogues and assessing the interlocutor's performance in the light of the standards for evaluating dialogue skills (self-esteem, good listening skills, expression of opinion, and respect for others). The instructor intervened only when one of the interlocutors deviated from the etiquette of dialogue or its topic. After the dialogue session, the instructor would hold a meeting to collect comments from all the students observing the interlocutors in terms of strengths, weaknesses, and opportunities for improvement and listen to the interlocutors' justification for their verbal and non-verbal behaviors during the dialogue. The instructor ended each session by providing feedback to the interlocutors. This explanation provides the answer to the first research question: "How can the role-play strategy be used in developing dialogue skills in a meaningful context for the advanced-level students in the Arabic Language Teaching Unit for Non-Native Speakers at Qassim University?".

## 4. Results and Discussion

The aim of this study was to identify the influence of role-play on the development of dialogue skills among advanced-level students studying Arabic as a second language. The theoretical and practical significance of the study stems from its usefulness for students, researchers, instructors, and curriculum planners. As per this study, role-playing would enhance the basic dialogue skills of advanced-level students who study Arabic as a second language, and instructors can use this strategy to motivate learning among students. The study provides ideas for researchers to carry out more scientific research investigating active learning strategies targeting students of Arabic as a second language. The study also offers a future vision for curriculum planners on the effectiveness of the role-play strategy in developing basic dialogue skills in Arabic.

In an endeavor to answer the second research question (are there any statistically significant differences between the scores of the experimental and control groups in the dialogue skill test due to the role-play strategy?), findings gleaned from the dialogue skills assessment scale were analyzed statistically, and the results are presented in Tables 2–8.

**Table 2.** Difference between the mean scores of students in the experimental and control groups on pre-administration of the dialogue skills assessment scale.

| No | Standards | Mean of Pre-Administration | |
| --- | --- | --- | --- |
| | | Experimental Group | Control Group |
| 1 | Self-esteem | 1.06 | 1.04 |
| 2 | Good listening | 2.06 | 1.09 |
| 3 | Expression of opinion | 1.06 | 1.05 |
| 4 | Respect for others | 1.08 | 1.06 |
| | Total | 7.06 | 6.04 |

**Table 3.** Statistical analysis of the differences between the mean scores of the students in the experimental and control groups on the pre-administration of the dialogue skills assessment scale.

| Group | N | Mean | St Dev | DF | Tabulated (t) | Sig. (2-Tailed) | Decision |
| --- | --- | --- | --- | --- | --- | --- | --- |
| Experimental | 25 | 7.06 | 2.32 | 24 | 1.711 | 0.85 | No statistically significant differences |
| Control | 25 | 6.04 | 2.16 | | | | |

Table 2 shows the scores of students in the experimental and control groups on the pre-administration of the dialogue skills assessment scale. The results reveal the weakness of the students in the experimental and control groups in terms of dialogue skills. The results also verify the equality of both groups at the beginning of the study, further confirmed by the statistical analysis of the differences between the mean scores of the students in the experimental and control groups on the pre-administration of the dialogue skills assessment scale, presented in Table 3.

**Table 4.** Difference between the mean scores of students in the experimental group in the pre–post administration of the dialogue skills assessment scale.

| No | Standards | Mean of Pre-Post Administration | |
|:---:|:---:|:---:|:---:|
| | | Pre-Administration | Post-Administration |
| 1 | Self-esteem | 1.06 | 4.06 |
| 2 | Good listening | 2.06 | 3.95 |
| 3 | Expression of opinion | 1.06 | 4.50 |
| 4 | Respect for others | 1.08 | 4.80 |
| | Total | 7.06 | 17.31 |

**Table 5.** Statistical analysis of the difference between the mean scores of the students in the experimental group on the pre- and post-administration of the dialogue skills assessment scale.

| Administration | N | Mean | St Dev | DF | Tabulated (t) | Sig. (2-Tailed) | Decision |
|:---:|:---:|:---:|:---:|:---:|:---:|:---:|:---:|
| Pre-test | 25 | 7.06 | 2.32 | | | | Extremely statistically significant differences |
| Post-test | 25 | 17.31 | 0.62 | 24 | 1.711 | 0.000 | |

**Table 6.** Difference between the mean scores of the students in the control group on the pre–post administration of the dialogue skills assessment scale.

| No | Standards | Mean of Pre-Post Administration | |
|:---:|:---:|:---:|:---:|
| | | Pre-Administration | Post-Administration |
| 1 | Self-esteem | 1.04 | 2.04 |
| 2 | Good listening | 1.09 | 2.01 |
| 3 | Expression of opinion | 1.05 | 3.02 |
| 4 | Respect for others | 1.06 | 2.02 |
| | Total | 6.04 | 9.09 |

**Table 7.** Statistical analysis of the difference between the mean scores of students in the control group on the pre- and post-administration of the dialogue skills assessment scale.

| Administration | N | Mean | St Dev | DF | Tabulated (t) | Sig. (2-Tailed) | Decision |
|:---:|:---:|:---:|:---:|:---:|:---:|:---:|:---:|
| Pre-test | 25 | 6.04 | 2.16 | | | | No statistically significant differences |
| Post-test | 25 | 9.09 | 1.96 | 24 | 1.711 | 0.19 | |

**Table 8.** Difference between the mean scores of the students in the experimental and control groups on the post-administration of the dialogue skills assessment scale.

| No | Standards | Mean of Post-Administration | |
|:---:|:---:|:---:|:---:|
| | | Experimental Group | Control Group |
| 1 | Self-esteem | 4.06 | 2.04 |
| 2 | Good listening | 3.95 | 2.01 |
| 3 | Expression of opinion | 4.50 | 3.02 |
| 4 | Respect for others | 4.80 | 2.02 |
| | Total | 17.31 | 9.09 |

Table 3 shows no statistically significant differences in the mean scores of students in the experimental and control groups on the pre-administration of the dialogue skills assessment scale. The pre–post administration of the dialogue skills assessment scale showed important results regarding the effect of role-play on improving the dialogue skills of the students in the experimental group, as illustrated in Table 4.

Table 4 shows a remarkable improvement in the average mean score of the students in the experimental group on pre- and post-administration of the dialogue skills assessment

scale. Although their average mean score was 7.06 on the pre-administration of the scale, it improved, to reach 17.31, upon post-administration. In addition, the pre–post scale results were treated statistically to prove the significance of the role-play strategy in the development of the dialogue skills of learners of Arabic as a second language at Qassim University, as illustrated in Table 5.

Table 5 shows extremely statistically significant differences in the mean scores of the students in the experimental group on the pre- and post-administration of the dialogue skills assessment scale in favor of post-administration. This result verifies the positive effect of the proposed role-play strategy on the dialogue skills of learners of Arabic as a second language at Qassim University. However, the results derived from the pre–post administration of the dialogue skills assessment scale did not show any remarkable differences in the scores of the control group, as shown in Table 6.

Table 6 shows little progress in the average mean score of the students in the control group on the pre- and post-administration of the dialogue skills assessment scale. Although their average mean score was 6.04 on the pre-administration of the scale, it improved slightly, to 9.09, upon post-administration. The pre–post scale results were also treated statistically to show the ineffectiveness of the traditional methods in enhancing the dialogue skills of learners of Arabic as a second language. In this context, Table 7 provides more information.

The results in Table 7 emphasize the need to introduce role-play as an effective method for enhancing dialogue skills since the traditional methods do not have any significant role in improving dialogue skills. Table 8 shows the difference between the mean scores of the students in the experimental and control groups on the post-administration of the dialogue skills assessment scale.

It is evident from the results shown in Table 8 that the scores of the experimental group were better than those of the control group in terms of dialogue skills, as the average of the experimental group scores on the scale post-administration was 17.31, while the average of the control group scores was 9.09 out of a total possible score of 20. This result emphasizes the effectiveness the role-play strategy in developing the dialogue skills of the advanced-level students of learners of Arabic as a second language in the Arabic Language Teaching Unit for Non-Native Speakers at Qassim University. In addition, the results of the post-administration of dialogue skills assessment scale of students of both experimental and control groups were treated statistically to verify the significance of the role-play strategy in the development of dialogue skills among learners of Arabic as a second language at Qassim University, as illustrated in Table 9.

**Table 9.** Statistical analysis of the difference between the mean scores of the students in the experimental and control groups on the post-administration of the dialogue skills assessment scale.

| Group | N | Mean | St Dev | DF | Tabulated (t) | Sig. (2-Tailed) | Decision |
|-------|---|------|--------|-----|---------------|-----------------|----------|
| Experimental | 25 | 17.31 | 0.62 | | | | Extremely statistically significant differences |
| Control | 25 | 9.09 | 1.96 | 24 | 1.711 | 0.000 | |

Table 9 shows tremendously statistically significant differences in the mean scores of the students in the experimental and control groups in favor of the students in the experimental group on the post-administration of the dialogue skills assessment scale. The results presented in Tables 8 and 9 prove the influence of the role-play strategy on the development of dialogue skills among the learners of Arabic as a second language at Qassim University. They also show slight differences in the development of the major dialogue skill standards, where respect for others came first, with an average of 4.80; the expression of opinion came second, with an average of 4.50; self-esteem came third, with an average of 4.06; and good listening came last, with an average of 3.95. The authors attribute these results to the effect of the role-play strategy, which helped the experimental

group to gain important dialogue skills related to self-esteem, good listening, expression of opinion, and respect for others. These results agree with those of previous research in that the role-play strategy contributes significantly to the development of dialogue skills among students by providing them an opportunity to use dialogue skills in active learning situations [12–14,26]. Role-play helps learners develop creativity by giving them an opportunity to learn in a relaxed learning environment. The study confirms that dialogue skills are crucial in language learning. Instructors should look for new ways to develop such skills to help learners express their opinions and support their arguments in a comfortable manner. In this regard, role-play allows the transfer of diverse academic and life experiences to the classroom. It allows students to practice roles that prepare them for a future life within the framework of a recreational activity according to an organized plan [3,14,22,27–29]. In addition, debates and arguments during role-play allow students to work in teams, to be good listeners, and to have a sense of respect toward others [9,19,20,23,30,31]. In this context, results of the study are consistent with those of previous research in highlighting that the role-play strategy allows students to explore and expand their network, to meet like-minded peers, to participate in healthy competition that could help build their character, and to enhance their ability to organize their thoughts to effectively put their argument forward. More importantly, students can also form informed arguments and use reasoning and evidence to support their standpoint [18,21,32–34].

This study revealed that using role-play in promoting discussion and dialogue is advantageous in many ways. It increases learners' ability to analyze information, describe situations, express opinions, explain reasons, discuss issues, persuade others, and make judgments. Role-play provides learners with the opportunity to experience the other side of the issue being discussed without being embarrassed or underestimated. Learners can comfortably act out new dialogues that mimic real-life situations but in a relaxed and active learning environment. The findings also show that role-play was effective in helping learners develop specific dialogue skills such as self-confidence, the ability to make decisions, and the determination to not give up easily. The students were able to show interest in what others say, ask for clarification of the content of the message instead of guessing, and make an effort to understand the point of view of others. The strategy facilitated the spontaneous flow of dialogue as students learned not to respond to or interrupt others before they finished their speech. Students were trained to focus on what is important in the opponent's speech; strive to provide convincing logical evidence that supports their point of view; adhere to the time specified for expressing their ideas; exchange opinions and ideas with the others to reach a solution to a controversial point; greet the opponent with kindness and affection before starting the dialogue; respect the speech and actions of others and not make fun of them; appreciate the talents and interests of others; avoid psychological tension and verbal violence; respect the opinions of others, even if they were opposed to their own opinions; and conclude the dialogue by shaking hands with the opponents. This clearly provides the answer to the third research question "What are the positive effects of the role-play strategy on dialogue skills in relation to self-esteem, respect for others, the expression of opinion, and good listening?" These results match similar research findings concluding that dialogue helps build positive relations between learners and instructors on the one hand and between them and their colleagues on the other hand, confirming mutual respect, acceptance of the other, and rejection of conflict. Dialogue helps to modify behavior and raise academic achievement, overcoming repression, and any slip of tongue is employed as a guide to positively support academic achievement [7,11,17,24,35–37]. However, the findings of this study disagree with the findings of a related study that showed that the role-play strategy is time-consuming, the behaviors of some students may be illogical and dissociated from reality, the strategy may not suit introvert students, and it may cause embarrassment and anxiety to those who have not mastered the art of acting. As per that study, role-play may become a source of entertainment, which invalidates learning [15]. These negative factors were carefully taken into consideration and avoided in this study through constant teacher supervision.

The role-play strategy was based on a variety of procedures. A group of students played real-life roles, and another group observed and took notes for strengths, weaknesses, and opportunities for improvement. The instructor monitored and directed in an atmosphere of cooperation toward achieving the prescribed teaching goal.

## 5. Delimitations

This study is delimited to developing the basic dialogue skills among the advanced level-six students in the first semester of the academic year 2022/2023 at the Arabic Language Teaching Unit for Non-Native Speakers at Qassim University. It is also delimited to 10 dialogue topics selected from the "Advanced Essay Course" for the sixth-level students (advanced) in the Language Teaching Unit for Non-Native Speakers at Qassim University, taught by a role-play strategy. These topics cover a variety of roles in social, environmental, economic, technological, educational, and daily-life situations where there may be a debate between two parties, such as an argument between a driver exceeding the speed limit and a police officer, a disagreement between a father and a son, negotiation between a customer and a salesman, conflict between a teacher and a student, and a debate between a tourist and a guide.

## 6. Conclusions and Implications

In conclusion, the authors attribute the development of learners' dialogue skills to the use of role-play that includes elements of suspense, fun, and attraction for students. The role-play method, as an active aspect of simulation, has proven successful in transferring the training to the trainees and has proven its effectiveness in the consolidation of concepts. Since the method is based on linking theory with practice of reality, it is more effective because training through practice is preferred over other types of education. The results of the study highlight the advantages of the role-play strategy in promoting dialogue skills among learners of Arabic as a second language at Qassim University, Saudi Arabia. The findings verify that role-play helps learners to use their experiences in real-life situations and develop the processes of thinking and analysis. It also creates a spirit of vitality and fun in the educational situation, strengthens positive communication, and develops social spirit and intimacy among students. The strategy helps identify students with outstanding and high abilities; addresses negative behaviors among students, such as introversion; provides the opportunity for practice, training, and feedback; provides students direct teaching experience; and develops self-confidence and pride among students.

In the light of the findings of this study, the authors recommend the following:

- Using multiple strategies and methods in teaching dialogue skills to learners of Arabic as a second language so that strategies that provide opportunities for positive interaction and participation are activated;
- Using role-play for developing some other language skills;
- Providing training programs for language instructors on the use of the role-play strategy in developing dialogue skills among learners of Arabic as a second language;
- Organizing workshops for instructors of Arabic as a second language on how to evoke previous experiences among students and how to link previous information with new information in Arabic language lessons;
- Developing a teacher's guide for teaching dialogue skills and including strategies and methods that have proven effective in teaching dialogue skills.

## 7. Suggestions for Further Research

The authors believe that the study may provide some useful ideas for other researchers to study innovative techniques for improving the acquisition of language skills among learners of Arabic as a second language. Examples of such ideas are as follows:

1. Evaluation of the teaching methods used in teaching dialogue skills to learners of Arabic as a second language;

2. The effect of a teaching method based on role-play in developing the poetry recitation skill among learners of Arabic as a second language;

3. The effect of a teaching method based on role-play in developing the speaking skill among learners of Arabic as a second language;

4. A study to identify the effect of different teaching methods (such as concept maps, mental visualization, and self-questioning) on the acquisition of dialogue skills among learners of Arabic as a second language.

**Author Contributions:** Methodology, M.S.A.-S.; Data curation, A.S.D.-A. All authors have read and agreed to the published version of the manuscript.

**Funding:** This research received no external funding.

**Institutional Review Board Statement:** The study was conducted in accordance with the Declaration of Helsinki, and approved by the Research Ethics Committees (REC) in the Arabic Language & English language & Translation Department, Qassim University (protocol code 21- Ara. -2022 &23-Eng-2021, 20 October 2022).

**Informed Consent Statement:** Informed consent was obtained from all subjects involved in the study.

**Data Availability Statement:** Not applicable.

**Conflicts of Interest:** The authors declare no conflict of interest.

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
