# Peer review of "The Effect of Role-Play on the Development of Dialogue Skills among Learners of Arabic as a Second Language"

_education, doi:10.3390/educsci13010050_

Round 1

Reviewer 1 Report

This study adopts an experimental approach to examine the effect of role-play on dialogue skills among advanced level Arabic learning students. The research design seems fine, but the whole article is poorly written. It seems that the author submitted the manuscript without carefully checking the whole text. Below are some aspects that the author must revise before the manuscript can be considered for publication.

1. The phrase “at Qassim University” in the title can be deleted to add the appeal of the research. The capitalization of the initial letters in the title is inconsistent, as some are not capitalized.

2. On Page 2 the heading “Problem of the study:” is rather weird. This makes the readers feel that the current study is “problematic”. No colon is needed at the end of the heading.

3. Three research questions are formulated in this study (Page 2). But in the next two sections (“purpose of the study” and “significance of the study”) the statements continue being numbered after the previous section. This is totally unacceptable. You must delete the automatic numbering in each paragraph in the “Purpose of the study” and “Significance of the study”.

4. It is rather odd to place “Definition of the terms” on Page 3 and 4. It would be much better to define the key terms in the Introduction section. I’m also not sure whether it is proper to put “Delimitations” in the current place (Page 3).

5. The purpose of Literature Review is to identify the research gap. Here I cannot see what research gap your study is going to fill.

6. In the research methods section, the research instrument used in the study, namely “Dialogue Skills Assessment Scale” must be elaborated. For instance, what are the exact statements in each indicator? How are the statements scored? Moreover, the process of the experiments also needs to be explained in detail. After reading this section, I have no idea what kind of role-play strategies were used in the experimental group.

7. In the Results and Discussion section, the first paragraph is still the explanation of the research methodology, thus it needs to be moved to the previous section.

8. On Page 12 the recommendations and suggestions are listed with bullet points, which does not conform to the conventions of academic journals.  

9. There are many grammatical problems with the English language, and some sentences are rather hard to understand. You must do a through proof-reading of the English language.   

Author Response

Authors' Reply to the Review Report (Reviewer 1)

No.

Reviewer 1 Comments

Authors’ Reply

1

The phrase “at Qassim University” in the title can be deleted to add the appeal of the research

This phrase is required by our institution for accreditation purposes.

Capitalization of the initial letters in the title is inconsistent, as some are not Capitalized.

All the necessary words are Capitalized.

2

“Problem of the study:” is rather weird.

This title is deleted and the section following it is added to the introduction.

No colon is needed at the end of the heading.

All colons at the end of headings are deleted.

3

You must delete the automatic numbering in each paragraph in the “Purpose of the study” and “Significance of the study”.

Automatic numbering, in each paragraph in the “Purpose of the study” and “Significance of the study”, is deleted .

4

 It is rather odd to place “Definition of the terms” on Page 3 and 4. It would be much better to define the key terms in the Introduction section.

All the key terms are defined in the introductory section.

I’m also not sure whether it is proper to put “Delimitations” in the current place (Page 3).

The “Delimitations” section has been moved to a proper place before the conclusion and implication section

5

The purpose of Literature Review is to identify the research gap. Here I cannot see what research gap your study is going to fill.

A paragraph that identifies the research gap has been added at the end of the Literature Review part.

6

In the research methods section, the research instrument used in the study, namely “Dialogue Skills Assessment Scale” must be elaborated. For instance, what are the exact statements in each indicator?

“Dialogue Skills Assessment Scale  has been elaborated and the exact statements in each indicator are provided .

The process of the experiments also needs to be explained in detail.

A section entitled “process of the experiment” is added in which detailed information about the implementation of the experiment is provided.

What kind of role-play strategies were used in the experimental group?

The “process of the experiment” section also, includes ten kinds of role-play strategies used in the experimental group. 

7

In the Results and Discussion section, the first paragraph is still the explanation of the research methodology, thus it needs to be moved to the previous section.

This paragraph has been moved to the previous section.

8

On Page 12 the recommendations and suggestions are listed with bullet points, which does not conform to the conventions of academic journals.  

All bullet points have been deleted and replaced with numbers.

There are many grammatical problems with the English language, and some sentences are rather hard to understand. You must do a thorough proof-reading of the English language.   

The whole manuscript has been revised by a native speaker working in Qassim University. Further editing will be done through the Journal.

Reviewer 2 Report

The paper presents the experimental study of using the role-play strategy to develop skills of dialogue in the university course of Arabic as a foreign language.

The study focuses on the effect of the role play teaching technique on developing of such skills of dialogue as self-esteem, active listening, expression of opinion and respect for others. To identify the effect of the role play the authors organized an experiment involving 50 university students of different origin studying Arabic as a foreign language. The participants were divided into a experimental and a control group and measured the four aforementioned skills of dialogue before and after the teaching intervention. The results of the study showed the significant improvement of the skills in the experimental group compared to the control group, which may be considered as reliable evidence of the effectiveness of the role-play method for developing the skills of dialogue.

The paper discusses a relevant topic which may be of significant importance for other researchers in the field of educational sciences, in particular for those interested in the teaching strategies aimed at developing universal communicative skills such as the skills of dialogue.

Yet, there is a number of questions which may help the authors to revise their manuscript.

1.     The introductory part of the manuscript is very extensive. It should be reduced and more focused. For example, the sections about significance and limitations of the study could be shifted to the Discussion section of the paper.

2.     In line 54 the authors inform about a pilot study. Who carried out the study? The authors or someone else?

3.     In the literature review the authors mention that the studies dealing with the use of the role-play method in developing the dialogue skills are not numerous at the regional and local levels. Yet, the use of the role-play in developing various communicative skills is widely discussed in the literature; therefore, it would be useful to approach to the issue more generally and to include relevant references discussing the same problem in EFL courses, for example.

4.     The definition of the term ‘strategy’ in general (p. 4) could be reduced to contain only the brief definition of “teaching strategy’ or omitted completely. The definition of the role-play strategy is sufficient to understand the topic.

5.     The definition of the dialogue skills needs to be more precise to differentiate two different understandings of the word “dialogue” – as language skills for verbal interaction (narrow understanding of the term) and as communicative skills involving the ability to effectively cooperate and interact with other people, i.e. the skills concerning the culture of dialogue (lines 169 – 176).

6.     The research design should be presented in more detail:

a)    The authors mention the pre-experimental orientation sessions to develop the experimental group participants’ awareness (line 279). How exactly was the awareness enhanced? Which learning tasks were used?

b)    How were the debates organized? Were there specific roles for each participant (for example proponent and opponent)?

7.     During the study four skills of dialogue were measured (line 266). It is recommended to describe the assessment procedure more precisely. How were the measured characteristics operationalized? Which quantitative indicators were measured to assess the four skills developed?

8.     In the Discussion section the authors state that the role-play method “has proven its effectiveness in the consolidation of concepts and the speed with which they are remembered by students”. (lines 457-458). Did the study also measure the development of cognitive skills such as conceptual development or memorizing? If yes, it should be mentioned in the Methods and Results sections. The same concerns the statement in the line 464: “to develop the processes of 464 thinking and analysis among them.”

9.     In line 93 one of the purposes of the study is stated: “Helping to reveal the differences in the cognitive domain between the average scores 93 of the students of the control group and the experimental group…”. Which differences in the cognitive domain does the study help to reveal?

10.  There is a number of grammatical mistakes, which should be corrected. For example, ‘the Arabic language instructor’(line 38); statistical significant (line 79); realistic-like (line 166) etc. Therefore, it would be useful to revise the manuscript to eliminate such mistakes.

11.  Reference 23 does not seem reliable.

12.  References 1, 2, 5, 10, 24 are unpublished references. It is recommended to provide some identification information for them (url, doi etc.).

Author Response

Authors' Reply to the Review Report (Reviewer 2)

No.

Reviewer 2 Comments

Authors’ Reply

1

The introductory part of the manuscript is very extensive. It should be reduced and more focused.

The introductory part has been reduced. The “Delimitations” section has been moved to a proper place before the conclusion and implication section. The purpose of the study section has been reduced and moved to the method section, Also, the section about significance of the study has been shifted to the Discussion section of the paper.

2

.     In line 54 the authors inform about a pilot study. Who carried out the study? The authors or someone else?

This problem has been solved by specifying who carried out the pilot study “Results of a pilot study, administered by the authors, revealed weaknesses in the language of dialogues used inside the classrooms.”

3

.     In the literature review the authors mention that the studies dealing with the use of the role-play method in developing the dialogue skills are not numerous at the regional and local levels. Yet, the use of the role-play in developing various communicative skills is widely discussed in the literature; therefore, it would be useful to approach to the issue more generally and to include relevant references discussing the same problem in EFL courses, for example.

This point has been clarified by adding more information to the paragraph “Although the use of the role-play in developing various communicative skills is widely discussed in the literature, yet the studies that dealt with the use of the role-play method in developing the dialogue skills for learners of Arabic as a second language- are very few at the regional and local levels.”

4

The definition of the term ‘strategy’ in general (p. 4) could be reduced to contain only the brief definition of “teaching strategy’ or omitted completely. The definition of the role-play strategy is sufficient to understand the topic.

This definition has been reduced and included in the introduction section.

5

The definition of the dialogue skills needs to be more precise to differentiate two different understandings of the word “dialogue” – as language skills for verbal interaction (narrow understanding of the term) and as communicative skills involving the ability to effectively cooperate and interact with other people, i.e. the skills concerning the culture of dialogue (lines 169 – 176).

Dialogue skills have been made more precise by adding the following sentence at the end of the provided definitions, “ In the current study, dialogue is approached extensively as communicative skills involving the ability to effectively cooperate and interact with other people”

6

The research design should be presented in more detail:

a)    The authors mention the pre-experimental orientation sessions to develop the experimental group participants’ awareness (line 279). How exactly was the awareness enhanced? Which learning tasks were used?

b)    How were the debates organized? Were there specific roles for each participant (for example proponent and opponent)?

A section entitled “process of the experiment” is added in which detailed information about the implementation of the experiment is provided. The purpose of each orientation session has been clearly stated and awareness enhancement procedures are provided. Explanation of how debates were organized and the roles of participant were also clarified.

7

During the study four skills of dialogue were measured (line 266). It is recommended to describe the assessment procedure more precisely. How were the measured characteristics operationalized? Which quantitative indicators were measured to assess the four skills developed?

“Dialogue Skills Assessment Scale  has been elaborated and the exact statements in each indicator have been provided .

8

 In the Discussion section the authors state that the role-play method “has proven its effectiveness in the consolidation of concepts and the speed with which they are remembered by students”. (lines 457-458). Did the study also measure the development of cognitive skills such as conceptual development or memorizing? If yes, it should be mentioned in the Methods and Results sections. The same concerns the statement in the line 464: “to develop the processes of 464 thinking and analysis among them.”

It is not the aim of the study to measure the development of memorizing. The study focuses on the development of the understanding and application of the introduced dialogue skills.

9

 In line 93 one of the purposes of the study is stated: “Helping to reveal the differences in the cognitive domain between the average scores 93 of the students of the control group and the experimental group…”. Which differences in the cognitive domain does the study help to reveal?

The study helps In revealing differences in the understanding and  application components of the cognitive domain.

10

There is a number of grammatical mistakes, which should be corrected. For example, ‘the Arabic language instructor’(line 38); statistical significant (line 79); realistic-like (line 166) etc. Therefore, it would be useful to revise the manuscript to eliminate such mistakes.

These mistakes have been fixed, and the whole manuscript has been revised by a native speaker working in Qassim University. Further editing will be done through the Journal.

11

Reference 23 does not seem reliable.

Reference 23 is deleted.

12

12.  References 1, 2, 5, 10, 24 are unpublished references. It is recommended to provide some identification information for them (url, doi etc.).

These references are written in Arabic and are available only in the university library database. All the available information is cited.

Round 2

Reviewer 1 Report

I'm happy with the revision, and the revised manuscript can be accepted for publication after checking again the English language.

Author Response

No.

Reviewer 1 Comments

Authors’ Reply

1.      

The revised manuscript can be accepted for publication after checking again the English language.

Authors requested use of one of the editing services provided by the journal. at https://www.mdpi.com/authors/english 

2.      

English language and style are fine/minor spell check required

Reviewer 2 Report

Dear authors,

thank you for your comment and corrections. Now the argument presented seems to be more logically coherent and clear. 

As to the edited manuscript, there are still several remarks to make:

1. The title of the paper should be edited. the phrase /at Quassim university/ narrows down the scale of your research to one specific institution. Presented in this way, it won't be interesting to the international audience. Moreover, the research should be quite general so that its design and findings could be useful for the researchers of other languages and in other institutions. 

2. The purpose of the study should have been left in the Introduction. Normally, the aims and purposes of the study are introduced in this section. It is too late to present them in the Methods section. 

3. In the paragraph describing the research gap it is reasonable to delete the sentence in lines 139-140. Such statement presents the gap as too narrow and specific. 

4. Several indicators in Table 1 should be reformulated to present the indicators as measurable or at least observable. For example, how can an observer find out that 'Interlocutor trusts his abilities'?

Author Response

No.

Reviewer 2 Comments

Authors’ Reply

1

The title of the paper should be edited. the phrase /at Qassim university/ narrows down the scale of your research to one specific institution. Presented in this way, it won't be interesting to the international audience. Moreover, the research should be quite general so that its design and findings could be useful for the researchers of other languages and in other institutions. 

The title has been edited. “The Effect of Using the Role-play Strategy on Developing the Dialogue Skills among Learners of Arabic as a Second Language ”

2

The purpose of the study should have been left in the Introduction. Normally, the aims and purposes of the study are introduced in this section. It is too late to present them in the Methods section. 

The purpose of the study has been added to the Introduction.

3

In the paragraph describing the research gap it is reasonable to delete the sentence in lines 139-140. Such statement presents the gap as too narrow and specific. 

This problem has been fixed. “The current study, therefore, intends to fill a gap in research by investigating the effect of the role-play strategy on improving four specific dialogue skills related to self-esteem, good listening, expression of opinion and respect for others, for non-native speakers studying Arabic as a second language. ”

4

Several indicators in Table 1 should be reformulated to present the indicators as measurable or at least observable. For example, how can an observer find out that 'Interlocutor trusts his abilities'?

Some indicators have been reformulated to be measurable. For example, indicator one “Interlocutor trusts his abilities.” has been adjusted as “Interlocutor shows confidence in his abilities.” Also, indicator two” Interlocutor does not find it difficult to make a decision.” has been reformulated “Interlocutor does not take much time to make a decision.”. Similarly, indicator eight and ten have also been modified to be “Interlocutor asks for clarification to understand others' point of view”; “Interlocutor takes notes of what is important in the opponent's speech”. Other indicators have also been reformulated.   

5

Moderate English changes required

Authors requested use of one of the editing services provided by the journal. at https://www.mdpi.com/authors/english